# DAMPs and RAGE Pathophysiology at the Acute Phase of Brain Injury: An Overview

**DOI:** 10.3390/ijms22052439

**Published:** 2021-02-28

**Authors:** Baptiste Balança, Laurent Desmurs, Jérémy Grelier, Armand Perret-Liaudet, Anne-Claire Lukaszewicz

**Affiliations:** 1Department of Neurological Anesthesiology and Intensive Care Medicine, Hospices Civils de Lyon, Hôpital Pierre Wertheimer, 69500 Bron, France; jeremy.grelier@gmail.com; 2Team TIGER, Lyon Neuroscience Research Centre, Inserm U1028, CNRS UMR 5292, 69500 Bron, France; 3Clinical Chemistry and Molecular Biology Laboratory, Hospices Civils de Lyon, Hôpital Pierre Wertheimer, 69500 Bron, France; laurent.desmurs@chu-lyon.fr (L.D.); armand.perret-liaudet@chu-lyon.fr (A.P.-L.); 4Team BIORAN, Lyon Neuroscience Research Centre, Inserm U1028, CNRS UMR 5292, 69500 Bron, France; 5Department of Neurological Anesthesiology and Intensive Care Medicine, Hospices Civils de Lyon, Hôpital Edouard Herriot, 69003 Lyon, France; anne-claire.lukaszewicz@chu-lyon.fr

**Keywords:** acute brain injuries, damage-associated molecular pattern molecules, receptor for advanced glycation end-products, biomarkers

## Abstract

Early or primary injury due to brain aggression, such as mechanical trauma, hemorrhage or is-chemia, triggers the release of damage-associated molecular patterns (DAMPs) in the extracellular space. Some DAMPs, such as S100B, participate in the regulation of cell growth and survival but may also trigger cellular damage as their concentration increases in the extracellular space. When DAMPs bind to pattern-recognition receptors, such as the receptor of advanced glycation end-products (RAGE), they lead to non-infectious inflammation that will contribute to necrotic cell clearance but may also worsen brain injury. In this narrative review, we describe the role and ki-netics of DAMPs and RAGE at the acute phase of brain injury. We searched the MEDLINE database for “DAMPs” or “RAGE” or “S100B” and “traumatic brain injury” or “subarachnoid hemorrhage” or “stroke”. We selected original articles reporting data on acute brain injury pathophysiology, from which we describe DAMPs release and clearance upon acute brain injury, and the implication of RAGE in the development of brain injury. We will also discuss the clinical strategies that emerge from this overview in terms of biomarkers and therapeutic perspectives

## 1. Introduction

The cells of the innate immune system search the extracellular environment for exogenous pathogens or self-molecules—either modified (e.g., oxidized lipids) or usually confined within cells via pattern recognition and scavenger receptors (PRRs) [1,2,3]. In the central nervous system these receptors are mainly expressed by microglial cells but also by astrocytes, neurons, endothelial cells, and infiltrating leukocytes upon vessel injury or blood–brain barrier (BBB) opening [1,3,4]. Nevertheless, resident microglial cells have a larger repertoire of Toll-like receptors (TLRs) and a greater amount of the receptor for advanced glycation end-products (RAGE) [1,5,6]. According to the context, PRRs may polarize resident microglial cells and infiltrating leukocytes toward a panel of pro- or anti-inflammatory phenotypes, playing a role in tissue damage clearance and repair but which may also trigger neuronal and glial degeneration [1,3,5,7,8]. Although acute brain injuries encompass a wide variety of mechanisms, they share a common feature as they all lead to acute cellular necrosis with the release of intracellular ions, adenosine di- or triphosphate (ATP), proteins, and nucleic acids in the extracellular space, and eventually the extravasation of red blood cells and hemoglobin [9,10,11,12]. Collectively, this wide variety of endogenous molecules released upon tissue injury are termed damage or danger-associated molecular patterns (DAMPs).

Neurological recovery following acute brain injury depends on the patient’s prior medical condition and primary injuries, but also on secondary damage [13,14]. Among other secondary insults, the inflammation that is triggered by the release of DAMPs plays a crucial role in the development of brain lesions which will be discussed in the present review. The levels of inflammatory markers released in the systemic blood, such as the C-reactive protein, or interleukin (IL) 6 and 12, are actually predictors of post-stroke neurological dysfunction [14,15,16]. Moreover, the concomitant occurrence of a systemic inflammation, such as during sceptic or anaphylactic shock, increases neurological damage at the acute phase of stroke [17].

The acute increase of extracellular potassium after ischemia or hemolysis leads to the depolarization and swelling of neighboring cells thereby starting propagating waves of spreading depolarization [18] (Figure 1A,B). Potassium and ATP release from dying cells also activate the inflammasome in neurons and astrocytes via pannexin1, purinoreceptor, and nucleotide oligomerization domain receptors (NOD-like receptors) [1,19]. Larger molecules, such as S100 proteins, hemoglobin derivatives or the high mobility group protein 1 (HMGB1), bind mainly to TLRs and RAGE [1,10]; the ligation of TLRs and RAGE triggers a series of cellular signaling events, including the activation of nuclear factor-kappa B (NF-κB), leading to the production of pro-inflammatory cytokines, and causing non-sceptic inflammation [1,20,21]. However, the consequences of DAMPs ligation to PRRs depends on the context and their concentration in the extracellular space. For instance, S100B, a calcium binding protein that has several intracellular actions in astrocytes, can promote cell growth in the nM extracellular range (i.e., physiological conditions) [22,23,24]; whereas at higher concentration S100B activates astrocytes with a pro-inflammatory phenotype and facilitate neuronal death [24,25]. HMGB1 expression may also promote neuroinflammation related to brain injury but the deletion of the encoding gene is not able to prevent such consequences [26,27]. Furthermore, sustained activation and up-regulation of RAGE on neurons has been reported to cause death via stimulating the production of reactive oxygen species [28], but also regulates neurite growth and cell survival [24], as well as apoptosis and autophagy [20,29,30]. However, some extracellular soluble truncated receptors, such as sRAGE, act as decoys for ligands, and thus have a cytoprotective effect against advanced glycation end-products (AGE) and RAGE interactions; serum levels of sRAGE have been investigated in pathological process and proposed as a biomarker of their intensity and severity of outcome [31]. It seems that the net effect of DAMPs and PRRs, such as RAGE, depends on the context, cell type, and the number of DAMPs and the level of expression of PRRs.

In this review, we aimed to describe the current knowledge of the implication of DAMPs and their receptors, in particular RAGE, in acute brain injury pathophysiology. We will also discuss the clinical strategies that emerge from this overview in terms of biomarkers and therapeutic perspectives.

## 2. Acute Lesion Progression Pathophysiology

### 2.1. Kinetics of DAMPs and Consequences after the Primary Insult

In addition to mechanical cell destruction, necroptosis is the main cell death pathway at the acute phase of traumatic brain injury (TBI), intracerebral or subarachnoid hemorrhage (SAH), and ischemic stroke (IS) [9,20,32]. Necroptosis is a morphologically lytic form of cell death implicating the receptor-interacting protein kinase 1 and 3 (RIPK1-RIPK3) and the mixed-lineage kinase domain-like pseudokinase (MLKL) pathway, resulting in the release of the contents of the cell into the extracellular space [33] (Figure 1A). The subsequent release of DAMPs peaks around 24 h after the primary insult and decreases thereafter over several days [4,12,20,34,35,36,37,38]. There is a spillover of DAMPs into the core of the primary injury, with a drastic decrease of intracellular HMGB1 [37] while HMGB1 is translocated from the nucleus to the cytoplasm of neurons but not glial cells at the periphery [4,9,37,39]. This extracellular release of HMGB1 may also act as a chemoattractant to monocytes [40] that infiltrate the core of the lesion from the first hours following the injury [4,37]. After several days, HMGB1-positive cells are mainly phagocytic microglial cells [4] (Figure 1D). Some authors have also described a biphasic expression of HMGB1 and S100 proteins with a second peak 14 days later [34,35]. The cellular consequences of DAMPs depend on the expression of PRRs but also on their post-translational modification. For instance, HMGB1 contains three highly conserved cysteines that are readily oxidized by reactive oxygen species, forming an intramolecular disulfide bridge thereby changing its conformation [41,42]. The reduced form released upon the primary injury binds the chemokine ligands (CXCL) 2 and 4 on monocytes and RAGE but not TLR-4, thereby promoting autophagy and acting as a chemoattractant for monocytes [40,43]. Conversely, the oxidized form may promote cell death and, at the late phase, vascular remodeling and progenitor cell migration via TLR signaling [35,43].

The expression of RAGE on neurons and microglial cells follows the same time course, with a biphasic pattern peaking on day 1 and 14 [12,20,34,35,36,44,45]. The amount of TLRs, which are predominantly expressed on microglial cells, increases at the subacute phase (i.e., after day 1) when activated microglia are present [34] (Figure 1D). The ligation of DAMPs to RAGE in turn activates the nuclear factor-kappa B (NF-κB) which in a positive feedback loop will increase the expression of RAGE. The activation of PRRs is related to different signaling pathways in neurons and glial cells. In neurons RAGE promotes the expression of proteins involved in necroptosis (i.e., MLKL) and autophagy (i.e., Becline-1); accordingly, blocking RAGE reduces autophagy, but also increases neuronal sensitivity to injury and apoptosis [20]. Later, at the subacute phase, TLRs and RAGE activation will polarize microglial cells and infiltrating leukocytes toward a pro-inflammatory phenotype, termed M1, with the release of cytokines such as tumor necrosis factor (TNF), IL-6 or IL-1 and the up-regulation of the inducible NO synthase [5,11,21,34,37,39]. Experimental overactivation of RAGE or TLRs by DAMPs injection or an increase in glucose degradation products, such as during hyperglycemia, will in turn increase the cytokine levels and worsen neuronal injuries [5,11,37]. The progression of secondary lesions follows different pathways. The ligation of TNF to the TNF receptor 1 (TNFR1) expressed by neurons will induce both necroptosis (via RIP1-3 and MLKL) and apoptosis (via caspase 3 and 8) [9,33]. Moreover, the pro-inflammatory cocktail can also activate astrocytes with a destructive phenotype (termed A1) leading to neuronal death and fewer synapses [7] (Figure 1C).

### 2.2. DAMPs Clearence

Under normal conditions, there is a continuous flow of cerebrospinal fluid (CSF) allowing the renewal of extracellular medium and the clearance of solutes such as amyloid-β (Aβ), also called the glymphatic system [46]. the subarachnoid CSF circulates in para-vascular spaces and enters the brain along penetrating arteries reaching the capillary bed; the interstitial fluid is then cleared along a perivenous drainage pathway and cervical lymphatic structures [46,47,48]. The glymphatic flux is driven by arterial pulse [47], water flux through aquaporin-4 expressed on astrocytes [46,48], as well as sleep cycles [30]. The CSF convection in the glymphatic system participates in the clearance of DAMPs to the peripheral blood [48], but also exhibits several changes following brain injury. The spreading depolarization that occurs at the very beginning of brain injuries triggers transient cell swelling, vasoconstriction [49,50] (Figure 1B), and increases the CSF flux in the glymphatic system, participating in the increase in brain water content [51]. However, the glymphatic flux is then impaired from 30 min to several days after the injury thereby leading to an accumulation of extracellular proteins such as DAMPs and Aβ [52,53,54,55]. DAMPs are also actively cleared from the extracellular space by infiltrating myeloid cells from the peripheral blood and activated resident glial cells. Infiltrating myeloid cells penetrate the lesion then differentiate into phagocytes that express scavenger receptors such as the macrophage scavenger receptor 1 (MSR1) that can bind and internalize HMGB1 and S100 proteins, as well as peridoxine [56]. These MSR1+ cells are present up to several days after the injury and exhibit an M2 phenotype, as opposed to the pro-inflammatory M1 phenotype of activated microglia that also internalize DAMPs [3,4,56]. Astrocytes are also able to internalize S100B into lysosomes by a RAGE-dependent mechanism [57]. Activated phagocytes may release HMGB1 from secretory lysosomes when modified lipids such as lysophosphatidylcholine are present at the subacute phase of acute brain injury, which may explain its biphasic release [58,59,60]. DAMPs can also traffic across the BBB by transcytosis in endothelial cells via RAGE and eventually reenter the brain from the blood compartment [61,62].

Increased expression of RAGE has been associated with the promotion of neuroinflammation in the main brain injuries, downstream HMGB1 release and NF-κB pathway activation [36]. Such a positive feedback loop of neuroinflammation may be blocked by intervention with antagonist of RAGE or the release of a truncated soluble form of RAGE (sRAGE) which may act as a decoy receptor to mitigate the inherent consequences of the inflammatory cascade. sRAGE is released upon brain injury [12], either secreted by astrocytes or monocytes [63] or from the cleavage of the membrane-bound RAGE [64]. sRAGE can scavenge extracellular and circulating DAMPs thus preventing their refilling into the brain [61] and reduce brain lesion progression [12,37,65]. The protective effect of recombinant sRAGE in cerebral parenchyma has been recently depicted in an animal model of SAH [65]. Although the characteristics of sRAGE sound interesting for clinical objectives, it has been little studied in brain injuries, with inconsistent results for outcomes such as in acute IS [66], or aSAH [31,67]. These discrepencies in data may result from different types of sRAGE studied and timing of measurements. We can speculate that the plasma level of sRAGE may be either positively associated with the intensity of injury, or negatively because of the consumption by its ligants (i.e., HMGB1 or S100B) possibly reflecting the healing process. Sometimes differences in plasma sRAGE may result from previous patient conditions such as diabetes or renal dysfunction [68]. In such a complex context it might be particularly interesting to combine more than one of these biomarker measurements to characterize the entire process.

## 3. A Biomarker Approach

A biomarker is a defined physiological characteristic that is measured and monitored objectively and reproducibly. It must be an indicator of biological processes or reactions to an exposure or an intervention, including therapeutic interventions. The ideal biomarker should be sensitive and specific to a pathological condition such as processes involved in early brain injury progression. The clearance of cerebral molecules from the extracellular space to the blood compartment as described in the previous section allows us to measure them from accessible biofluids such as CSF, serum, plasma, and/or whole blood [47,48,69].

A biomarker is considered to be clinically useful if it provides information in addition to the clinical assessment; it can shorten the time to diagnosis or therapeutic decision, or allow a therapeutic follow-up if the kinetics of change are rapid and if the concentration in biological fluid is correlated with the intensity of the disease [70,71]. The biomarker and its analytical method must be adapted to the needs of physicians to shorten the time of a therapeutic decision. Physicians face several challenges at the acute phase of brain injury. The type and severity of brain injury (e.g., ischemic of hemorrhagic) is hard to diagnose when access to brain imaging is not possible, such as during out of hospital triage or in pediatric emergencies when general anesthesia could be required to perform cerebral imaging [72,73,74,75]. The early prognostication of future complication and long-term outcome is also uncertain using clinical data. For instance, neurological score and radiological assessment are not always reliable at the early phase of TBI, IS, and stroke mimics such as migraine or epileptic seizures [76,77]. However, they remain the gold standard to guide clinical decision. During this early phase of brain injury management, several applications of DAMPs as biomarkers have been investigated, such as early injury severity rating [78], distinguishing ischemic or hemorrhagic lesions [79], or prognostication of long-term outcome [80].

The most extensively studied biomarkers are those associated with damage to neurons (neuron-specific enolase, NSE) and glial cells (S100B and glial fibrillary acidic protein—GFAP, mainly expressed in astrocytes) [81]. S100B and GFAP allow the diagnosis of intracranial hemorrhage in patients with cerebral ischemia [78,79,80,82,83,84,85,86,87,88] (Table 1). They are also accurate diagnostic biomarkers that facilitate the triage and the need for brain imaging within a few hours following mild and moderate TBI [89,90,91] (Table 2). Moreover, NSE and S100B have demonstrated a prognostic potential on early and long-term functional recovery after ischemic or hemorrhagic strokes and TBI [80,84,85,86,87,88,92,93,94,95]. Likewise, elevated HMGB1 and sRAGE are predictors of poor prognosis after an IS [67,96] (Table 1 and Table 2). Changes in HMGB1, S100B, and sRAGE concentrations during the hospital stay can also help to diagnose delayed injury such as proximal vasospasm and delayed cerebral ischemia after SAH [10,31,97] (Table 1). Nevertheless, concomitant extracerebral injuries may bias the interpretation of DAMPs and sRAGE levels as they induce a similar increase [98]. S100B is also located extracerebrally, such as in chondrocytes, Schwann cells, adipocytes and malignant melanoma cells, which influences S100B serum levels [81]. As described in Table 1 and Table 2, there is a large variability in the sensitivity and specificity of DAMPs and sRAGE. This is likely to be in relation to the complexity of acute brain injury. Therefore, a single protein biomarker may not be sufficient, and it is reported that a panel of several biomarkers of different cellular origins and different kinetic profiles makes it possible to better predict the prognosis [99,100] and improve the diagnostic accuracy [101,102,103] (Table 3).

Unfortunately, there is no standardized assays measuring these biomarkers, which constitutes a barrier to the dissemination of these measures in routine clinical practice with established and reproducible cutoffs. For example, S100B can be measure with several techniques such as enzyme-linked immunosorbent assay (ELISA; Sangtec, Saluggia Italy, canAG Diagnostics, Gothenburg Sweden), automated luminometric immunoassay (Diasorin, sangtec, Italy) or automated electrochemiluminescence immunoassay (Cobas, Roche Diagnostics, Mannheim Germany); each method has its own normal and pathological ranges [104,105]. Pre-analytical steps could also be a source of heterogeneity, although S100B measurement have been reported to be unaffected by hemolysis [106], storing, changes in temperature or freeze-thaw cycles [107]. Depending on the clinical and analytics challenges one technique should be preferred over the others. For instance, when rapid decision making is more relevant, rapid quantitative immunoassays are faster and available at any time compared to the traditional ELISA assays (i.e., less than one hour vs. 4 to 5 h). Some biomarkers are at very low concentration in biofluids with sensitivity issues; therefore, new platforms using digital ELISA (e.g., Simoa, Quanterix, Billerica USA) have been developed to overcome this limitation and greatly improved the sensitivity of the tests (e.g., Tau protein, GFAP or Neurofilaments) [108].

## 4. Therapeutic Perspectives

The brain levels of DAMPs and RAGE have been used as indirect markers of anti-inflammatory drug efficacy following experimental acute brain injury [112,113], but these could also be therapeutic targets, either by blocking the receptors leading to cellular death or by improving clearance of DAMPs from the brain. For instance, blocking RAGE (e.g., with *N*-benzyl-4-chloro-*N*-cyclohexylbenzamide, or glycyrrhizin that can be used in humans) can reduce cerebral expression of PRRs, HMGB1 release, and cerebral edema in models of TBI [39,114] or SAH [115,116]. Reducing cerebral edema following acute brain injury can be critical in cases of intracranial hypertension that worsens the prognosis [117]. However, while reducing cerebral edema and expression of PPRs, blocking RAGE might also increase neuronal apoptosis and down-regulate autophagy thereby worsening neuronal lesions [20]. Autophagy may serve as a major cytoprotective process by maintaining cellular homeostasis and recycling cytoplasmic contents [118]. Its role after acute brain injury remains controversial [119,120], and the safety and efficacy of a strategy resulting in autophagy down-regulation remains to be evaluated. Another cell death pathway involves the release of TNF that binds the TNFR1 leading to both necroptosis and apoptosis [9,33]. The intracellular recruitment of RIPK1 by TNFR1 starts a cascade that leads to cell death. Interestingly, the recruitment of RIPK1 is blocked by the endogenous TNF alpha-induced protein 3 (also known as A20) [9]. The level of A20 therefore represents an endogenous regulator of inflammation-induced cell death. Recently some authors described that the expression of A20 can be increased by the administration of melatonin, thereby reducing cell death and cerebral edema in experimental TBI or SAH models [9,121,122]. Melatonin is already approved for the treatment of primary insomnia in adults to promote sleep and could be a potent therapeutic agent to be used at the acute and subacute phase of brain injury. The effect of daily melatonin administration in the acute phase of stroke on inflammation (i.e., IL-6 systemic levels) is currently under investigation in a randomized controlled trial (NCT03843008).

The BBB forms a natural shield that prevents neuroprotective drugs from accessing brain tissue. However, small lipophilic molecules can cross the BBB and were therefore used to design nanoparticles carrying hydrophilic drugs to deliver them to the brain tissue [123]. The intranasal administration of nanoparticles can accumulate their cargo in the brain unlike systemic free administration [124]. Moreover, some nanoparticles release their cargo only in an acidic environment, which is a characteristic of ischemic tissue at risk of irreversible damage; this molecular design allows the enriching of acutely injured tissue without the risk of unfavorable off-target effects [125]. These new technological breakthroughs are a major step forward in specifically targeting injured cerebral tissue with drugs that interfere with DAMPs and PPRs. For instance, nanoparticles have been used to carry antibodies against TNF or IL-6, as well as short hairpin ribonucleic acid silencing the expression of TLRs thereby attenuating inflammation and improving tissue recovery [126,127].

The clearance of DAMPs is also critical to resolve brain inflammation. As described in the previous section the cleaning of DAMPs from the brain involves their scavenging by infiltrating monocytes and microglial cells, and their removal via CSF outflow or drainage. Infiltrating monocytes and macrophages can scavenge DAMPs via several receptors, such as MSR1 [56]. The expression of scavenger receptors is under the regulation of several transcription factors; MafB regulates the expression of MSR1. Vitamin A also has profound effects on cell proliferation, homeostasis, and metabolism through nuclear retinoid receptors [128]. Recently, agonists of the retinoid X receptor have been reported to induce MafB mRNA expression in macrophages [129]. Tamibarotene is an orally active, synthetic retinoid agonist that is more active than all-trans retinoic acid and improves the survival of patients with acute promyelocytic leukemia [130]. In an experimental stroke model tamibarotene enhanced expression of both MafB and MSR1, and in turn reduced the infarct volume [56]. It could therefore be a future therapeutic agent for acute brain injury; however, its tolerance and efficacy remain to be investigated in patients with acute brain injuries. Nevertheless, there is currently a phase 2 clinical trial in patients with Alzheimer’s disease investigating the administration of tamibarotene (NCT01120002). sRAGE can also act as a decoy receptor and scavenge DAMPs in the extracellular space and blood compartment. The administration of recombinant sRAGE reduced cell death in vitro [12,65] and at high dosage in in vivo experimental models of IS and SAH [12,37,65]. However, its efficacy and tolerance in humans is so far unknown. In addition, restoring the glymphatic system circulation after acute brain injury to enhance DAMPs clearance to the peripheral blood is also a future approach that should be investigated. CSF flux through the glymphatic system can be influenced by sleep or sedation. For instance, some data suggest that sevoflurane can enhance the clearance of Aβ by up-regulating the expression of aquaporin-4 in astrocytes thereby improving the CSF flux in the glymphatic system [69]. As low doses of sevoflurane are not supposed to increase the cerebral blood volume, this could be an additional tool to restore CSF clearance after acute brain injury and needs to be evaluated [131]. Once evacuated to the blood compartment, DAMPs can be further cleared using a hemadsorption extracorporeal circulation with porous polymer beads or nanotraps, thereby removing them from the whole body and preventing their refilling to the brain [132,133]. This strategy has yielded promising results in pre-clinical models of sepsis or TBI [133,134]. CSF can also be evacuated by an external ventricular drain (EVD), either when there is a hydrocephalus or to monitor intracranial pressure [135]; it can be used to measure DAMPs following acute brain injury [55], but has yet to be evaluated for the clearance of DAMPs. EVDs are also being used in several clinical trials investigating the effect of intraventricular fibrinolysis to restore the CSF circulation on patient outcome after SAH, when the subarachnoid space is covered with blood (e.g., the FIVHeMA trial, NCT03187405).

## 5. Materials and Methods

The methodological approach is a narrative review to summarize the main published findings in the field of DAMPs, RAGE, and acute brain injuries [116]. We convened a working group of experts in neurological intensive care medicine (B.B. and J.G.) and molecular biology and clinical chemistry (L.D. and A.P.L.). Section 2 is based on a searched of the MEDLINE database using the PubMed search engine, looking for original articles published in English language using the keywords: “Receptor for Advanced Glycation End-products”, “Damage-associated molecular patterns”, “traumatic brain injury”, “subarachnoid hemorrhage” and “stroke”; with the following Boolean operators: ((Receptor for Advanced Glycation End-products[Title/Abstract]) OR (Damage-associated molecular patterns[Title/Abstract]) OR (S100[Title/Abstract])) AND ((traumatic brain injury[Title/Abstract]) OR (subarachnoid hemorrhage[Title/Abstract]) OR (stroke[Title/Abstract])) NOT (Review[Publication Type]). We then excluded reviews and articles not related to acute brain injuries and selected articles reporting data on acute brain injury pathophysiology. We also searched clinicaltrails.gov for unpublished studies described in Section 4.

## 6. Conclusions

To conclude, the release of DAMPs upon acute brain injury triggers RAGE activation and steril inflammation, which is a major component of secondary damage progression. It promotes cerebral edema, triggers secondary vascular lesions such as vasospasm and microthrombi, and leads to cellular death via the release of inflammatory cytokines and the polarization of microglial cells toward an M1 phenotype. DAMPs are then cleared from the extracellular space through the glymphatic system to the peripheral blood, and also scavenge by the inflitration of circulating leukocytes followed by activated mciroglia. The concentration of DAMPs in the peripheral blood are therefore good biomarkers of the severity and type of the primary injury. DAMPs can also be used to predict the outcome and monitor the course of brain injury. The detection of a combination of different DAMPs may also improve their diangositic performance. However, the proinflamatory cytokine cocktail that has already been released upon the primary injury may worsen cerebral lesions, and targeting the downstream intracellular consequences could be a future therapeutic strategy. The recent developpment of nanoparticles may, in the near future, allow the administration of neuroprotective drugs specifically in the injured area where in may be most helpful. Clearing and scanvenging DAMPs from brain injury seems also to be critical to prevent lesion progression, and could repesent future theraptic targets.

## Figures and Tables

**Figure 1 ijms-22-02439-f001:**
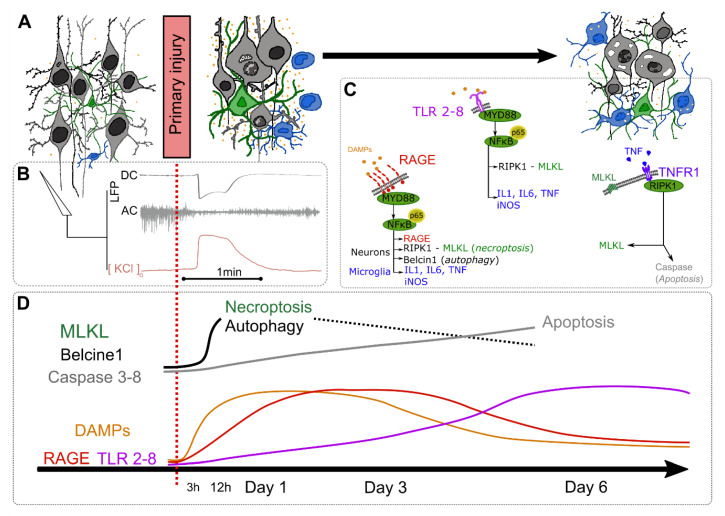
Damage-associated molecular patterns (DAMPs) and pattern-recognition receptors (PRRs) changes at the acute phase of brain injury. (**A**) Morphological changes of neurons (grey), astrocytes (green), microglial and infiltrating leukocytes (blue), and DAMPs release (orange); (**B**) local field potential (LFP) and extracellular KCl recordings during the spreading depolarization triggered by the primary injury: the direct current (DC; 0–0.5Hz) shift is associated with a decrease of neuronal activity (AC; >0.5 Hz) and a KCl release; (**C**) Pattern-recognition receptors (PRRs) activation pathways; (**D**) kinetics of DAMPs and PRR expression as well as the course of cell death mechanisms. DAMPs: Damage-associated molecular patterns; IL: Interleukin; iNOS: inducible nitric oxide synthase; LFP: local field potential; MLKL: Mixed-lineage kinase domain-like pseudokinase; MyD88: Myeloid differentiation primary response 88; NFκB: nuclear factor-kappa B; RAGE: Receptor for advanced glycation end-products; RIPK1: receptor-interacting protein kinase 1; TLR: Toll-like receptor; TNF: Tumor necrosis factor; TNFR1: Tumor necrosis factor receptor 1.

**Table 1 ijms-22-02439-t001:** Biomarker approach of damage-associated molecular patterns following ischemic or hemorrhagic stroke.

Author (Year)	Marker	Study Design	Applications	Outcome	Results
Ren et al. [79] (2016)	GFAP	Case-control (132 IS; 57 controls)	Diagnosis	Stroke subtype	GFAP discriminated IS from ICH within 4.5 h of symptoms onset (Se = 61%, Sp = 96%, AUC = 0.86).
Luger et al. [82] (2020)	GFAP	Prospective observational (251 IS)	Diagnosis	Stroke subtype	ICH patients had higher serum level of GFAP than IS patients and mimics.
			Clinical severity	CT lesion volume	GFAP was correlated with ICH volume (r = 0.296).
Katsanos et al. [83] (2017)	GFAP	Case-control (191 IS; 79 controls)	Diagnosis	Stroke subtype	GFAP discriminated IS from ICH (Se = 91%, Sp = 97%, AUC = 0.97).
			Clinical severity	NIHSS	No correlation has been found between serum levels of GFAP and stroke severity on admission in IS of different subtype.
Zhou et al. [84] (2016)	S100B	Prospective observational (46 ICH; 71 IS)	Diagnosis	Stroke subtype	ICH patients had higher plasma level of S100B than IS patients.
			Clinical severity	NIHSS	Positive correlation between S100B and infarct size (r = 0.820).
			Prognosis	90-day mRS	S100B predicted a poor prognosis (Se = 100%, Sp = 76%, AUC = 0.92).
Balança et al. [78] (2020)	S100B	Prospective observational (81 SAH)	Clinical severity	3-day GOS	Severe EBI was associated with higher S100B concentration at admission or day 1 (Cliff’s delta = 0.73, 95% CI [0.46; 0.88]), which predicted early recovery (AUC = 0.87).
Branco et al. [85] (2018)	S100B	Prospective observational (131 IS)	Prognosis	12-week upper limb functioning	S100B predicted hand functioning (Se = 69%, Sp = 90%, AUC = 0.84).
Kedziora et al. [86] (2020)	S100B	Prospective observational (55 SAH)	Prognosis	GOS at ICU discharge	S100B predicted ICU outcome (Se = 91%, Sp = 63%, AUC = 0.81).
Kellermann et al. [92] (2016)	S100B	Prospective observational (45 SAH)	Prognosis	6-month GOS	S100N at day 1 predicted poor outcome (OR = 4.38, 95% CI, [1.08; 17]). A negative correlation was found between serum level of S100B and 6 months GOS (r = 0.434).
Kiiski et al. [87] (2018)	S100B; NSE	Prospective observational (47 SAH)	Prognosis	6-month mRS	No correlation has been found between biomarker concentrations and the neurological outcome.
Abboud et al. [88] (2018)	S100B; NSE	Prospective observational (52 SAH)	Prognosis	6-month GOS	S100B and NSE at day 1 predicted good outcome with 100% specificity.
Quintard et al. [80] (2015)	S100B; NSE	Prospective observational (48 SAH)	Prognosis	6-month GOS	Poor neurological outcome was predicted by S100B levels at day 5 (AUC = 0.91) and NSE level at day 7 (AUC = 0.83).
Aida et al. [31] (2019)	sRAGE	Prospective observational (627 SAH)	Complication	symptomatic vasospasm	sRAGE level was lower in symptomatic vasospasm group on day 7, and predicted poor outcome (Se = 70%, Sp = 86%, AUC = 0.77).
Yang et al. [67] (2018)	sRAGE	Case-control (108 SAH and 108 controls)	Prognosis	6-month GOS score	sRAGE within 24 h after SAH was associated with clinical severity and poor 6-month outcomes (Se = 83%, Sp = 75%, AUC = 0.82).
Tang et al. [66] (2015)	sRAGE	Case-control (106 IS and 150 controls)	Prognosis	3-month mRS	sRAGE level was higher in the IS group and predicted poor neurological score (OR = 2.44, 95% CI [1.16; 5.16]).
Tsukagawa et al. [109] (2017)	HMGB1	Case-control (183 IS and 16 controls)	Prognosis	1-year mRS	HMGB1 level on admission was a significant independent predictor of poor outcome (OR = 2.34, 95% CI [1.02; 5.38]).
Wang et al. [96] (2020)	HMGB1	Prospective observational (132 IS)	Prognosis	3-month mRS	High concentration of HMGB1 at 6 h after thrombolytic therapy was associated with poor outcome (Se = 87%, Sp = 74%, AUC = 0.87).
Kiiski et al. [110] (2017)	HMGB1	Prospective observational (47 SAH)	Prognosis	6-month mRS	No correlation has been found between biomarker concentrations and the neurological outcome.

ICH: intracerebral hemorrhage; IS: ischemic stroke; SAH: subarachnoid hemorrhage; NIHSS: national institute of health stroke scale; ICU: intensive care unit; GFAP: Glial fibrillary acidic protein; NSE: neuron-specific enolase; GOS: Glasgow outcome scale; mRS; modified Rankin scale; CT: computed tomography; CI: confidence interval; AUC: area under the curve; OR: odds ratio; PV: predictive value; Se: sensitivity; Sp: specificity; CVS: cerebral vasospasm subtype; sRAGE: soluble receptor for advanced glycation end-products; HMGB1: high mobility group box 1.

**Table 2 ijms-22-02439-t002:** Biomarker approach of damage-associated molecular patterns following traumatic brain injury.

Author (Year)	Marker	Study Design	Applications	Outcome	Results
Osier et al. [111] (2018)	S100B SNP in genes	Prospective study (305 severe TBI)	Risk factor	3, 6, 12, 24-months GOS	The variant allele of one S100B SNP (rs1051169) was identified as a protective factor at 3 months (OR = 0.39), 6 months (OR = 0.34), 12 months (OR = 0.32) and 24 months (OR = 0.30).
Mahan et al. [89] (2019)	GFAP; S100B	Case-control (118 TBI and 37 controls)	Diagnosis	CT lesion (CT+)	These biomarkers were significantly higher in patients CT+ than patients CT−. GFAP had the greatest prognostic capacity (0–8 h: AUC = 0.89; and 12–32 h: AUC = 0.94).
Meier et al. [90] (2017)	GFAP; S100B	Case-control (32 TBI and 29 controls)	Diagnosis	Sport-related concussion	S100B could predict concussion in athletes (AUC = 0.72) but not for GFAP.
Çevik et al. [91] (2019)	GFAP; S100B	Cross-sectional study (48 mild TBI)	Diagnosis	CT lesion (CT+)	S100B and GFPA were significantly higher in mild TBI patients with CT+.
Kellermann et al. [92] (2016)	S100B	Prospective observational (57 TBI)	Prognosis	6-month GOS	S100B at day 1 predicted a poor outcome (OR = 7.6, 95% CI [2.24; 25.80]). A negative correlation was found between serum level of S100B and 6 months GOS (r = 0.494).
Park et al. [94] (2018)	S100B; NSE; IL-6	Prospective observational (15 pediatric patients with TBI)	Prognosis	6-month GOS	S100B and NSE correlated with the severity of brain injury and predicted poor neurological outcome, but not IL6.
Park et al. [93] (2019)	S100B; NSE	Prospective observational (10 pediatric patients with TBI)	Prognosis	6-month GOS	High concentration of S100B at 1 week after admission was associated with poor outcome. No statistical difference was found for NSE.
Thelin et al. [95] (2016)	S100B; NSE	Prospective observational (417 TBI)	Prognosis	3-month GOS	CSF level of S100B at day 1 predicted poor outcome (OR = 4.15, 95% CI [1.34; 12.84]).
				6-month GOS	S100B had a betted diagnostic accuracy than NSE.

GOS: Glasgow outcome scale; GFAP: glial fibrillary acidic protein; NSE: neuron-specific enolase; TBI: trauma brain injury; CT: computed tomography; CI: confidence interval; AUC: area under the curve; OR: odds ratio; SNP: single nucleotide polymorphism; Se: specificity; Se: sensitivity.

**Table 3 ijms-22-02439-t003:** Biomarkers panels approach.

Author (Year)	Marker	Study Design	Applications	Outcome	Results
Posti et al. [101] (2020)	Aβ40; Aβ42; GFAP; H-FABP; IL-10; NF-L; S100β; Tau	Prospective observational (136 TBI patients)	Diagnosis	CT lesion (CT+)	The combination of two biomarkers (Aβ40 and IL-10) and clinical characteristics (HCTS) were better to discriminate CT+ and CT− patients (Se = 91%, Sp = 59%) than HCTS alone (Se = 97%, Sp = 22%).
Posti et al. [102] (2019)	Aβ40; Aβ42; GFAP; H-FABP; IL-10; NF-L; S100β; Tau	Prospective observational (160 TBI including 93 mild TBI)	Diagnosis	CT lesion (CT+)	H-FABP + S100B + Tau were the best association to diagnose CT+ in mild TBI (Se = 100%, Sp = 46%) whereas it was a question of the combination of GFAP + H-FAB + IL-10 in group with all severities TBI (Se = 100%, Sp = 39%).
Chen et al. [103] (2019)	S100β; NSE; hK6; PGDS	Case-control (10 severe TBI and 10 controls)	Diagnosis	Subtype severe TBI	S100B/hK6 and NSE/PGDS ratios (both AUC = 1.00) diagnosed severe TBI with greater accuracy than S100B and NSE alone (both AUC = 0.97).
Thelin et al. [99] (2019)	S100β; NSE; GFAP; Tau; NF-L; UCH-L1	Prospective observational (172 TBI)	Prognosis	12-month GOS	GFAP + NF-L was the best association to predict poor outcome.
Di Battista et al. [100] (2015)	S100β; GFAP; NSE; BDNF; MCP-1; ICAM-5; PRDX-6	Prospective observational (85 TBI)	Prognosis	Death and 6-month GOS	S100B and GFAP levels were the most discriminating to predict poor outcome (AUC = 0.82 and AUC = 0.71, respectively) and mortality (AUC = 0.86 and AUC = 0.79, respectively).

IL: interleukin; GOS: Glasgow outcome scale; GFAP: glial fibrillary acidic protein; UCH-L1: ubiquitin C-terminal hydrolase L1; NSE: neuron-specific enolase; TBI: trauma brain injury; CT: computed tomography; CI: confidence interval; AUC: area under the curve; OR: odds ratio; Aβ: β-amyloid; hK6: human kallikrein 6; PGDS: prostaglandin D2 synthase; H-FABP: heart fatty-acid bonding protein; NF-L: neurofilament light; ICAM: intercellular adhesion molecule; PRDX: peroxiredoxin; BDNF: brain-derived neurotrophic factor; MCP: monocyte chemoattractant protein; HCTS: Helsinki computerized tomography score; Sp: specificity; Se: sensitivity.

## Data Availability

Not applicable.

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
