# Peer review of "DAMPs and RAGE Pathophysiology at the Acute Phase of Brain Injury: An Overview"

_ijms, 2021, doi:10.3390/ijms22052439_

Round 1

Reviewer 1 Report

The authors have written a review of the role and importance of DAMPs and RAGE in the progression of brain injuries. They have introduced the topic explaining the major role of the various key players and their effects long term.  The review is interesting, but an overhaul of the english is required. Throughout the article the use of plural and singular is  not correct as well as the use of articles being missing. Also, the final section of the review gives very few examples of how these targets could be used to limit disease progression. This seems underwhelming however as there is vast literature regarding targeting these both directly or with nanomedicines etc. This section should be expanded at least to cover the various methods and research pathways that are looking into this.

Author Response

We thank you for the work that has been done while reviewing our manuscript which has been improved. Please find below our reply to your remarks. The changes have been highlighted in red in the revised version.

Point 1: The review is interesting, but an overhaul of the english is required. Throughout the article the use of plural and singular is not correct as well as the use of articles being missing.

Response 1: The manuscript has been revised by a native speaker from our institution publication department.

Point 2: Also, the final section of the review gives very few examples of how these targets could be used to limit disease progression. This seems underwhelming however as there is vast literature regarding targeting these both directly or with nanomedicines etc. This section should be expanded at least to cover the various methods and research pathways that are looking into this.

Response 2: The reviewer is right to highlight that other neuroprotective strategies have been investigated to improve neurological recovery after an acute brain injury. Some of them does not specifically target DAMPS or RAGE and we chose not to implement them in this last section because we think that they are not in the scope of the herein review. We originally wanted to discuss strategies that could be implemented in the clinical practice in a near future, however we agree with the reviewer that some advances were made in DAMPs clearance and drug administrations that should be discussed.

We enriched the final section with a paragraph about nanoparticles, as suggested by the reviewer, which represents a major breakthrough in drug delivery that may also be implemented in future clinical trials and may overcome some past negative results. We also added a discussion about DAMPs clearance from the blood using hemadsorption that is under investigation in other pathological condition and could be used after acute brain injuries. Both changes are highlighted in red in the revised manuscript.

Reviewer 2 Report

DAMPs, such as S100B, participate in the regulation of cell growth and survival but they trigger cellular damage as their concentration increases in the extracellular space. Authors  searched the MEDLINE database for 21
‘DAMPs’ or ‘RAGE’ or ‘S100B’ and ‘traumatic brain injury’ or ‘subarachnoid hemorrhage’ or ‘stroke’. Authors  selected original articles reporting data on acute brain injury pathophysiology, from which we describe DAMP release and clearance upon acute brain injury, and the implication of 24
RAGE in the development of brain injury. Authors  also discussed the clinical strategies that emergefrom this overview in terms of biomarkers and therapeutic perspective

This is well conducted review article

I have only  minor comments to do:

Authors should add some further information about the role of DAMPS in stroke pathogenesis on their introduction section

Authors should expand their conclusion section 

Authors should add on conclusion a sentence about the role of inflammation on stroke and other vascular complication and they should add on their reference section these citations about this issue: 

  • Tuttolomondo A, Maida C, Pinto A. Diabetic foot syndrome: Immune-inflammatory features as possible cardiovascular markers in diabetes. World J Orthop. 2015 Jan 18;6(1):62-76
  • Pinto A, Tuttolomondo A, Di Raimondo D, Fernandez P, Licata G. Risk factors profile and clinical outcome of ischemic stroke patients admitted in a Department of Internal Medicine and classified by TOAST classification. Int Angiol. 2006 Sep;25(3):261-7 
  • Shi K, Tian DC, Li ZG, Ducruet AF, Lawton MT, Shi FD. Global brain inflammation in stroke. Lancet Neurol. 2019 Nov;18(11):1058-1066
  • Systemic inflammatory challenges compromise survival after experimental stroke via augmenting brain inflammation, blood- brain barrier damage and brain oedema independently of infarct size.Dénes A, Ferenczi S, Kovács KJ.J Neuroinflammation. 2011 Nov 24;8:164.

Author Response

We thank you for the work that has been done while reviewing our manuscript which has been improved. Please find below our reply to your remarks. The changes have been highlighted in red in the revised version.

Point 1: Authors should add some further information about the role of DAMPS in stroke pathogenesis on their introduction section.

Response 1: A paragraph reporting the role of inflammation in stroke has been added to the introduction section. This section includes 3 of the 4 references that the reviewers asked us to implement to our manuscript. We did not include the reference reviewing the pathophysiology of the diabetic foot syndrome, as it is not in the scope of the herein review.

Point 2: Authors should expand their conclusion section. Authors should add on conclusion a sentence about the role of inflammation on stroke and other vascular complication.

Response 2: The conclusion section has been expended accordingly echoing the introduction section. Therefore, a sentence has been added concluded on the role of inflammation during stroke and delayed vascular complication during subarachnoid hemorrhage and traumatic brain injuries.

Round 2

Reviewer 1 Report

that authors have adequetely revised all requested sections and have improved the paper to a publishable form.